# How many degrees of freedom do we need to train deep networks: a loss landscape perspective

**Brett W. Larsen**[1], **Stanislav Fort**[1], **Nic Becker**[1], **Surya Ganguli**[1,2]
[1]Stanford University, [2]Meta AI | Correspondence to: `bwlarsen@stanford.edu`

## Abstract

A variety of recent works, spanning pruning, lottery tickets, and training within random subspaces, have shown that deep neural networks can be trained using far fewer degrees of freedom than the total number of parameters. We analyze this phenomenon for random subspaces by first examining the success probability of hitting a training loss sublevel set when training within a random subspace of a given training dimensionality. We find a sharp phase transition in the success probability from 0 to 1 as the training dimension surpasses a threshold. This threshold training dimension increases as the desired final loss decreases, but decreases as the initial loss decreases. We then theoretically explain the origin of this phase transition, and its dependence on initialization and final desired loss, in terms of properties of the high-dimensional geometry of the loss landscape. In particular, we show via Gordon's escape theorem, that the training dimension plus the *Gaussian width* of the desired loss sublevel set, projected onto a unit sphere surrounding the initialization, must exceed the total number of parameters for the success probability to be large. In several architectures and datasets, we measure the threshold training dimension as a function of initialization and demonstrate that it is a small fraction of the total parameters, implying by our theory that successful training with so few dimensions is possible precisely because the Gaussian width of low loss sublevel sets is very large. Moreover, we compare this threshold training dimension to more sophisticated ways of reducing training degrees of freedom, including lottery tickets as well as a new, analogous method: lottery subspaces.

## 1 Introduction

How many parameters are needed to train a neural network to a specified accuracy? Recent work on two fronts indicates that the answer for a given architecture and dataset pair is often much smaller than the total number of parameters used in modern large-scale neural networks. The first is successfully identifying lottery tickets or sparse trainable subnetworks through iterative training and pruning cycles (Frankle & Carbin, 2019). Such methods utilize information from training to identify lower-dimensional parameter spaces which can optimize to a similar accuracy as the full model. The second is the observation that constrained training within a random, low-dimension affine subspace is often successful at reaching a high desired train and test accuracy on a variety of tasks, provided that the training dimension of the subspace is above an empirically-observed threshold training dimension (Li et al., 2018). These results, however, leave open the question of why low-dimensional training is so successful and whether we can theoretically explain the existence of a threshold training dimension.

In this work, we provide such an explanation in terms of the high-dimensional geometry of the loss landscape, the initialization, and the desired loss. In particular, we leverage a powerful tool from high-dimensional probability theory, namely Gordon's escape theorem, to show that this threshold training dimension is equal to the dimension of the full parameter space minus the squared Gaussian width of the desired loss sublevel set projected onto the unit sphere around initialization. This theory can then be applied in several ways to enhance our understanding of neural network loss landscapes. For a quadratic well or second-order approximation around a local minimum, we derive an analytic bound on this threshold training dimension in terms of the Hessian spectrum and the distance of the initialization from the minimum. For general models, this relationship can be used in reverse to measure important high-dimensional properties of loss landscape geometry. For example, by

performing a tomographic exploration of the loss landscape, i.e. training within random subspaces of varying training dimension, we uncover a phase transition in the success probability of hitting a given loss sublevel set. The threshold training dimension is then the phase boundary in this transition, and our theory explains the dependence of the phase boundary on the desired loss sublevel set and the initialization, in terms of the Gaussian width of the loss sublevel set projected onto a sphere surrounding the initialization.

Motivated by lottery tickets, we furthermore consider training not only within random dimensions, but also within optimized subspaces using information from training in the full space. Lottery tickets can be viewed as constructing an optimized, axis-aligned subspace, i.e. where each subspace dimension corresponds to a single parameter. What would constitute an optimized choice for general subspaces? We propose two new methods: burn-in subspaces which optimize the offset of the subspace by taking a few steps along a training trajectory and lottery subspaces determined by the span of gradients along a full training trajectory (Fig. 1). Burn-in subspaces in particular can be viewed as lowering the threshold training dimension by moving closer to the desired loss sublevel set. For all three methods, we empirically explore the threshold training dimension across a range of datasets and architectures.

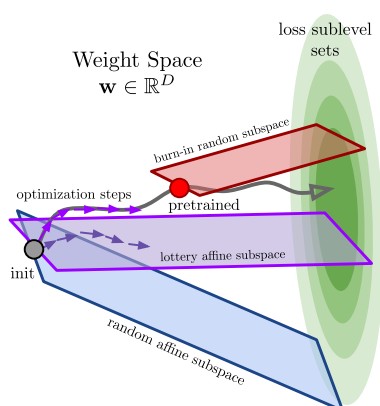

Figure 1: Illustration of finding a point in the intersection between affine subspaces and low-loss sublevel set. We use three methods: 1) *random affine subspaces* (blue) containing the initialization, 2) *burn-in affine subspaces* (red) containing a pre-trained point on the training trajectory, and 3) *lottery subspaces* (purple) whose span is defined by the steps of a full training trajectory.

**Related Work:** An important motivation of our work is the observation that training within a random, low-dimensional affine subspace can suffice to reach high training and test accuracies on a variety of tasks, provided the training dimension exceeds a threshold that was called the *intrinsic dimension* by (Li et al., 2018) and which is closely related to what we call the threshold training dimension. Li et al. (2018) however did not explore the dependence of this threshold on the quality of the initialization or provide a theoretical explanation for its existence. Our primary goal then is to build on this work by providing such an explanation in terms of the geometry of the loss landscape and the quality of initialization. Indeed understanding the geometry of high-dimensional error landscapes has been a subject of intense interest in deep learning, see e.g. Dauphin et al. (2014); Goodfellow et al. (2014); Fort & Jastrzebski (2019); Ghorbani et al. (2019); Sagun et al. (2016; 2017); Yao et al. (2018); Fort & Scherlis (2019); Papyan (2020); Gur-Ari et al. (2018); Fort & Ganguli (2019); Papyan (2019); Fort et al. (2020), or Bahri et al. (2020) for a review. But to our knowledge, the Gaussian width of sublevel sets projected onto a sphere surrounding initialization, a key quantity that determines the threshold training dimension, has not been extensively explored in deep learning.

Another motivation for our work is contextualizing the efficacy of diverse more sophisticated network pruning methods like lottery tickets (Frankle & Carbin, 2019; Frankle et al., 2019). Further work in this area revealed the advantages obtained by pruning networks not at initialization (Frankle & Carbin, 2019; Lee et al., 2018; Wang et al., 2020; Tanaka et al., 2020) but slightly later in training (Frankle et al., 2020), highlighting the importance of early stages of training (Jastrzebski et al., 2020; Lewkowycz et al., 2020). We find empirically, as well as explain theoretically, that even when training within random subspaces, one can obtain higher accuracies for a given training dimension if one starts from a slightly pre-trained or burned-in initialization as opposed to a random initialization.

## 2 AN EMPIRICALLY OBSERVED PHASE TRANSITION IN TRAINING SUCCESS

We begin with the empirical observation of a phase transition in the probability of hitting a loss sublevel set when training within a random subspace of a given training dimension, starting from some initialization. Before presenting this phase transition, we first define loss sublevel sets and two different methods for training within a random subspace that differ only in the quality of the initialization. In the next section, we develop theory for the nature of this phase transition.

**Loss sublevel sets.** Let $\hat{\mathbf{y}} = f_{\mathbf{w}}(\mathbf{x})$ be a neural network with weights $\mathbf{w} \in \mathbb{R}^D$ and inputs $\mathbf{x} \in \mathbb{R}^k$. For a given training set $\{\mathbf{x}_n, \mathbf{y}_n\}_{n=1}^N$ and loss function $\ell$, the empirical loss landscape is given by $\mathcal{L}(\mathbf{w}) = \frac{1}{N} \sum_{n=1}^N \ell\Big(f_{\mathbf{w}}(\mathbf{x}_n), \mathbf{y}_n\Big)$. Though our theory is general, we focus on classification for our experiments, where $\mathbf{y} \in \{0, 1\}^C$ is a one-hot encoding of $C$ class labels, $\hat{\mathbf{y}}$ is a vector of class probabilities, and $\ell(\hat{\mathbf{y}}, \mathbf{y})$ is the cross-entropy loss. In general, the loss sublevel set $S(\epsilon)$ at a desired value of loss $\epsilon$ is the set of all points for which the loss is less than or equal to $\epsilon$:

$$S(\epsilon) := \{\mathbf{w} \in \mathbb{R}^D : \mathcal{L}(\mathbf{w}) \leq \epsilon\}. \tag{2.1}$$

**Random affine subspace.** Consider a $d$-dimensional random affine hyperplane contained in $D$-dimensional weight space, parameterized by $\boldsymbol{\theta} \in \mathbb{R}^d$: $\mathbf{w}(\boldsymbol{\theta}) = \mathbf{A}\boldsymbol{\theta} + \mathbf{w}_0$. Here $\mathbf{A} \in \mathbb{R}^{D \times d}$ is a random Gaussian matrix with columns normalized to 1 and $\mathbf{w}_0 \in \mathbb{R}^D$ a random weight initialization by standard methods. To train within this subspace, we initialize $\boldsymbol{\theta} = \mathbf{0}$, which corresponds to randomly initializing the network at $\mathbf{w}_0$, and we minimize $\mathcal{L}(\mathbf{w}(\boldsymbol{\theta}))$ with respect to $\boldsymbol{\theta}$.

**Burn-in affine subspace.** Alternatively, we can initialize the network with parameters $\mathbf{w}_0$ and train the network in the full space for some number of iterations $t$, arriving at the parameters $\mathbf{w}_t$. We can then construct the random burn-in subspace

$$\mathbf{w}(\boldsymbol{\theta}) = \mathbf{A}\boldsymbol{\theta} + \mathbf{w}_t, \tag{2.2}$$

with $\mathbf{A}$ chosen randomly as before, and then subsequently train within this subspace by minimizing $\mathcal{L}(\mathbf{w}(\boldsymbol{\theta}))$ with respect to $\boldsymbol{\theta}$. The random affine subspace is identical to the burn-in affine subspace but with $t = 0$. Exploring the properties of training within burn-in as opposed to random affine subspaces enables us to explore the impact of improving the initialization, via some information from the training data, on the success of subsequent restricted training.

**Success probability in hitting a sublevel set.** In either training method, achieving $\mathcal{L}(\mathbf{w}(\boldsymbol{\theta})) = \epsilon$ implies that the intersection between our random or burn-in affine subspace and the loss sublevel set $S(\epsilon')$ is non-empty for all $\epsilon' \geq \epsilon$. As both the subspace $\mathbf{A}$ and the initialization $\mathbf{w}_0$ leading to $\mathbf{w}_t$ are random, we are interested in the success probability $P_s(d, \epsilon, t)$ that a burn-in (or random when $t = 0$) subspace of training dimension $d$ actually intersects a loss sublevel set $S(\epsilon)$:

$$P_s(d, \epsilon, t) \equiv \mathbb{P}\Big[S(\epsilon) \cap \big\{\mathbf{w}_t + \mathrm{span}(\mathbf{A})\big\} \neq \emptyset\Big]. \tag{2.3}$$

Here, $\mathrm{span}(\mathbf{A})$ denotes the column space of $\mathbf{A}$. Note in practice we cannot guarantee that we obtain the minimal loss in the subspace so we use the best value achieved by Adam (Kingma & Ba, 2014) as an approximation. Thus, the probability of achieving a given loss sublevel set via training constitutes an approximate lower bound on the probability in (2.3) that the subspace actually intersects the loss sublevel set.

**Threshold training dimension as a phase transition boundary.** We will find that for any fixed $t$, the success probability $P_s(d, \epsilon, t)$ in the $\epsilon$ by $d$ plane undergoes a sharp phase transition. In particular for a desired (not too low) loss $\epsilon$, it transitions sharply from 0 to 1 as the training dimension $d$ increases. To capture this transition we define:

**Definition 2.1.** *[Threshold training dimension] The threshold training dimension $d^*(\epsilon, t, \delta)$ is the minimal value of $d$ such that $P_s(d, \epsilon, t) \geq 1 - \delta$ for some small $\delta > 0$.*

For any chosen criterion $\delta$ (and fixed $t$) we will see that the curve $d^*(\epsilon, t, \delta)$ forms a phase boundary in the $\epsilon$ by $d$ plane separating two phases of high and low success probability. This definition also gives an operational procedure to approximately measure the threshold training dimension: run either the random or burn-in affine subspace method repeatedly over a range of training dimensions $d$ and record the lowest loss value $\epsilon$ found in the plane when optimizing via Adam. We can then construct the empirical probability across runs of hitting a given sublevel set $S(\epsilon)$ and the threshold training dimension is lowest value of $d$ for which this probability crosses $1 - \delta$ (where we employ $\delta = 0.1$).

## 2.1 AN EMPIRICAL DEMONSTRATION OF A TRAINING PHASE TRANSITION

In this section, we carry out this operational procedure, comparing random and burn-in affine subspaces across a range of datasets and architectures. We examined 3 architectures: 1) `Conv-2`

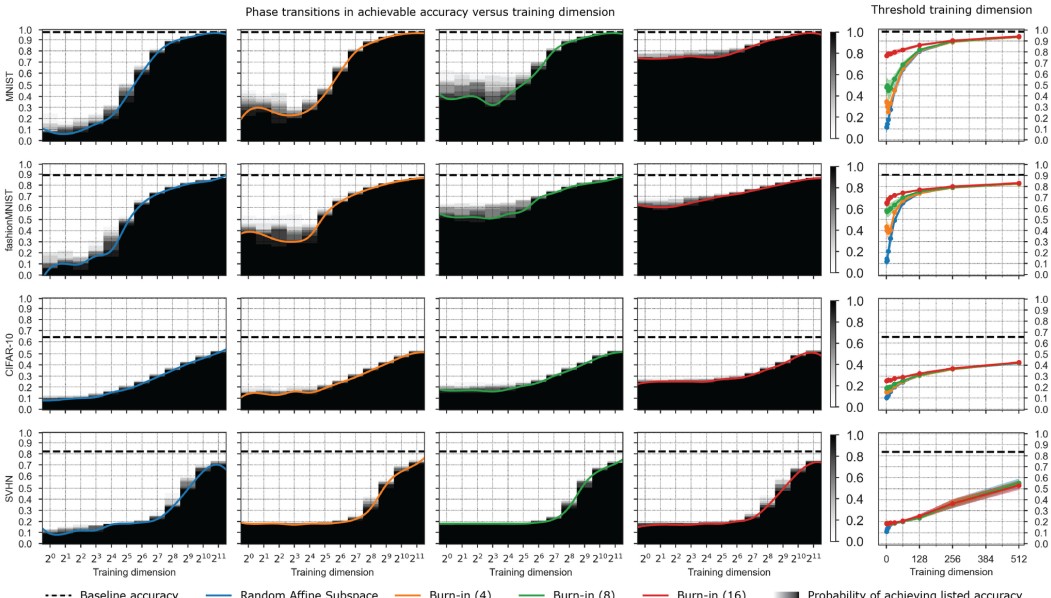

Figure 2: An empirical phase transition in training success on 4 datasets (4 rows) for a `Conv-2` comparing random affine subspaces (column 1) and burn-in affine subspaces with $t = 4, 8, 16$ burn-in steps (columns 2,3,4). The black-white color maps indicate the empirically measured success probability $P_s(d, \epsilon, t)$ in (2.3) in hitting a training loss sublevel set (or more precisely a training accuracy super-level set). This success probability is estimated by training on 10 runs at every training dimension $d$ and burn-in time $t$. The horizontal dashed line represents the baseline accuracy obtained by training the full model for the same number of epochs. The colored curves indicate the threshold training dimension $d^*(\epsilon, t, \delta)$ in definition 2.1 for $\delta = 0.1$. The threshold training dimensions for the 4 training methods are copied and superimposed in the final column.

which is a simple 2-layer CNN with 16 and 32 channels, ReLU activations and `maxpool` after each convolution followed by a fully connected layer; 2) `Conv-3` which is a 3-layer CNN with 32, 64, and 64 channels but otherwise identical setup to `Conv-2`; and 3) `ResNet20v1` as described in He et al. (2016) with on-the-fly batch normalization (Ioffe & Szegedy, 2015). We perform experiments on 5 datasets: MNIST (LeCun et al., 2010), Fashion MNIST (Xiao et al., 2017), CIFAR-10 and CIFAR-100 (Krizhevsky et al., 2014), and SVHN (Netzer et al., 2011). Baselines and experiments were run for the same number of epochs for each model and dataset combination; further details on architectures, hyperparameters, and training procedures are provided in the appendix. The code for the experiments was implemented in JAX (Bradbury et al., 2018).

Figure 2 shows results on the training loss for 4 datasets for both random and burn-in affine subspaces with a `Conv-2`. We obtain similar results for the two other architectures (see Appendix). Figure 2 exhibits several broad and important trends. First, for each training method within a random subspace, there is indeed a sharp phase transition in the success probability $P_s(d, \epsilon, t)$ in the $\epsilon$ (or equivalently accuracy) by $d$ plane from 0 (white regions) to 1 (black regions). Second, the threshold training dimension $d^*(\epsilon, t, \delta)$ (with $\delta = 0.1$) does indeed track the tight phase boundary separating these two regimes. Third, broadly for each method, to achieve a lower loss, or equivalently higher accuracy, the threshold training dimension is higher; thus one needs more training dimensions to achieve better performance. Fourth, when comparing the threshold training dimension across all 4 methods on the same dataset (final column of Figure 2) we see that at high accuracy (low loss $\epsilon$), *increasing* the amount of burn in *lowers* the threshold training dimension. To see this, pick a high accuracy for each dataset, and follow the horizontal line of constant accuracy from left to right to find the threshold training dimension for that accuracy. The first method encountered with the lowest threshold training dimension is burn-in with $t = 16$. Then burn-in with $t = 8$ has a higher threshold training dimension and so on, with random affine having the highest. Thus the main trend is, for some range of desired accuracies, burning *more* information into the initialization by training on the training data *reduces* the number of subsequent training dimensions required to achieve the desired accuracy.

Figure 3 shows the threshold training dimension for each train and test accuracy level for all three models on MNIST, Fashion MNIST and CIFAR-10. The broad trends discussed above hold robustly for both train and test accuracy for all three models.

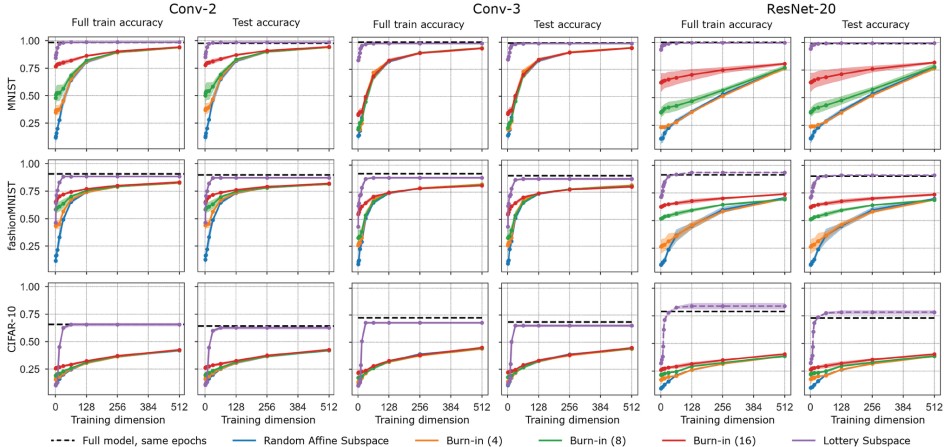

Figure 3: The threshold training dimension $d^*(\epsilon, t, \delta)$ in definition 2.1. Here we focus on small dimensions and lower desired accuracies to emphasize the differences in threshold training dimension across different training methods. The purple curves are generated via a novel lottery subspace training method which we introduce in section 4. The curves summarize data for 10 runs for Conv-2, 5 runs for Conv-3, and 3 runs for ResNet20; the choice of $\delta$ will determine how many runs must successfully hit the sublevel set when reading off $d^*$. The dimensions of the full parameter space for the experiments with CIFAR-10 are 25.6k for Conv-2, 66.5k for Conv-3, and 272.5k for ResNet20. On the other two datasets, the full parameter space is 20.5k for Conv-2, 61.5k for Conv-3, and 272.2k for ResNet20. The black dotted line is the accuracy obtained by training the full model for the same number of epochs.

## 3 A THEORY OF THE PHASE TRANSITION IN TRAINING SUCCESS

Here we aim to give a theoretical explanation for the major trends observed empirically above, namely: (1) there exists a phase transition in the success probability $P_s(d, \epsilon, t)$ yielding a phase boundary given by a threshold training dimension $d^*(\epsilon, t, \delta)$; (2) at fixed $t$ and $\delta$ this threshold increases as the desired loss $\epsilon$ decreases (or desired accuracy increases), indicating more dimensions are required to perform better; (3) at fixed $\epsilon$ and $\delta$, this threshold decreases as the burn-in time $t$ increases, indicating *fewer* training dimensions are required to achieve a given performance starting from a *better* burned-in initialization. Our theory will build upon several aspects of high-dimensional geometry which we first review. In particular we discuss, in turn, the notion of the Gaussian width of a set, then Gordon's escape theorem, and then introduce a notion of local angular dimension of a set about a point. Our final result, stated informally, will be that the threshold training dimension plus the local angular dimension of a desired loss sublevel set about the initialization must equal the total number of parameters $D$. As we will see, this succinct statement will conceptually explain the major trends observed empirically. First we start with the definition of Gaussian width:

**Definition 3.1** (Gaussian Width). *The Gaussian width of a subset $S \subset \mathbb{R}^D$ is given by (see Figure 4):*

$$w(S) = \frac{1}{2} \mathbb{E} \sup_{\mathbf{x}, \mathbf{y} \in S} \langle \mathbf{g}, \mathbf{x} - \mathbf{y} \rangle, \quad \mathbf{g} \sim \mathcal{N}(\mathbf{0}, \mathbf{I}_{D \times D}).$$

As a simple example, let $S$ be a solid $l_2$ ball of radius $r$ and dimension $d \ll D$ embedded in $\mathbb{R}^D$. Then its Gaussian width for large $D$ is well approximated by $w(S) = r\sqrt{d}$.

**Gordon's escape theorem.** The Gaussian width $w(S)$ of a set $S$, at least when that set is contained in a unit sphere around the origin, in turn characterizes the probability that a random subspace intersects that set, through Gordon's escape theorem (Gordon, 1988):

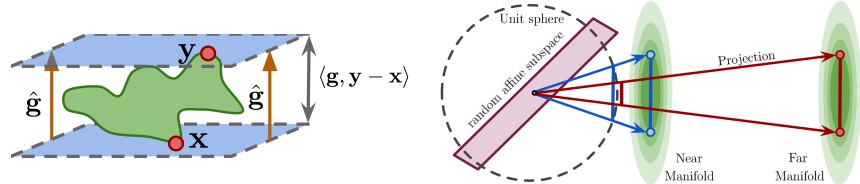

Figure 4: **Left panel:** An illustration of measuring the width of a set $S$ (in green) in a direction $\hat{\mathbf{g}}$ by identifying $\mathbf{x}, \mathbf{y} \in S$ in $\max_{\mathbf{x},\mathbf{y} \in S} \hat{\mathbf{g}} \cdot (\mathbf{y} - \mathbf{x})$. The expectation of this width using random vectors $\mathbf{g} \sim \mathcal{N}(\mathbf{0}, \mathbf{I}_{D \times D})$ instead of $\hat{\mathbf{g}}$ is twice the Gaussian width $w(S)$. Intuitively, it is the characteristic extent of the set $T$ over all directions rescaled by a factor between $D/\sqrt{D+1}$ and $\sqrt{D}$. **Right panel:** Illustration of projecting manifolds on the unit sphere and Gordon's escape theorem. The same manifold far from the sphere will have a smaller projection to it than the one that is close, and therefore it will be harder to intersect with an affine subspace.

**Theorem 3.1.** *[Escape Theorem] Let $S$ be a closed subset of the unit sphere in $\mathbb{R}^D$. If $k > w(S)^2$, then a $d = D - k$ dimensional subspace $Y$ drawn uniformly from the Grassmannian satisfies (Gordon, 1988):*

$$\mathbb{P}\Big(Y \cap S = \emptyset\Big) \geq 1 - 3.5 \exp\left[-\left(k/\sqrt{k+1} - w(S)\right)^2/18\right].$$

A clear explanation of the proof can be found in Mixon (2014).

Thus, the bound says when $k > w^2(S)$, the probability of no intersection quickly goes to $1 - \epsilon$ for any $\epsilon > 0$. Matching lower bounds which state that the intersection occurs with high probability when $k \leq w(S)^2$ have been proven for spherically convex sets (Amelunxen et al., 2014). Thus, this threshold is sharp except for the subtlety that you are only guaranteed to hit the spherical convex hull of the set (defined on the sphere) with high probability.

When expressed in terms of the subspace dimension $d = D - k$, rather than its co-dimension $k$, these results indicate that a $d$-dimensional subspace will intersect a closed subset $S$ of the unit sphere around the origin with high probability if and only if $d + w(S)^2 \geq D$, with a sharp transition at the threshold $d^* = D - w(S)^2$. This is a generalization of the result that two random subspaces in $\mathbb{R}^D$ of dimension $d$ and $d_2$ intersect with high probability if and only if $d + d_2 > D$. Thus we can think of $w(S)^2$ as playing a role analogous to dimension for sets on the centered unit sphere.

### 3.1 INTERSECTIONS OF RANDOM SUBSPACES WITH GENERAL SUBSETS

To explain the training phase transition, we must now adapt Gordon's escape theorem to a general loss sublevel set $S$ in $\mathbb{R}^D$, and we must take into account that the initialization $\mathbf{w}_t$ is not at the origin in weight space. To do so, we first define the projection of a set $S$ onto a unit sphere centered at $\mathbf{w}_t$:

$$\text{proj}_{\mathbf{w}_t}(S) \equiv \{(\mathbf{x} - \mathbf{w}_t)/||\mathbf{x} - \mathbf{w}_t||_2 : \mathbf{x} \in S\}. \tag{3.1}$$

Then we note that any affine subspace $Y$ of the form in eq. (2.2) centered at $\mathbf{w}_t$ intersects $S$ if and only if it intersects $\text{proj}_{\mathbf{w}_t}(S)$. Thus we can apply Gordon's escape theorem to $\text{proj}_{\mathbf{w}_t}(S)$ to compute the probability of the training subspace in eq. (2.2) intersecting a sublevel set $S$. Since the squared Gaussian width of a set in a unit sphere plays a role analogous to dimension, we define:

**Definition 3.2** (Local angular dimension). *The local angular dimension of a general set $S \subset \mathbb{R}^D$ about a point $\mathbf{w}_t$ is defined as*

$$d_{\text{local}}(S, \mathbf{w}_t) \equiv w^2(\text{proj}_{\mathbf{w}_t}(S)). \tag{3.2}$$

An escape theorem for general sets $S$ and affine subspaces now depends on the initialization $\mathbf{w}_t$ also, and follows from the above considerations and Gordon's original escape theorem:

**Theorem 3.2.** *[Main Theorem] Let $S$ be a closed subset of $\mathbb{R}^D$. If $k > w(\text{proj}_{\mathbf{w}_t}(S))^2$, then a $d = D - k$ dimensional affine subspace drawn uniformly from the Grassmannian and centered at $\mathbf{w}_t$ satisfies:*

$$\mathbb{P}\Big(Y \cap S = \emptyset\Big) \geq 1 - 3.5 \exp\left[-\left(k/\sqrt{k+1} - w(\text{proj}_{\mathbf{w}_t}(S))\right)^2/18\right].$$

To summarise this result in the context of our application, given an arbitrary loss sublevel set $S(\epsilon)$, a training subspace of training dimension $d$ starting from an initialization $\mathbf{w}_t$ will hit the (convex hull) of the loss sublevel set with high probability when $d + d_{\text{local}}(S(\epsilon), \mathbf{w}_t) > D$, and will miss it (i.e have empty intersection) with high probability when $d + d_{\text{local}}(S(\epsilon), \mathbf{w}_t) < D$. This analysis thus establishes the existence of a phase transition in the success probability $P_s(d, \epsilon, t)$ in eq. (2.3), and moreover establishes the threshold training dimension $d^*(\epsilon, t, \delta)$ for small values of $\delta$ in definition 2.1:

$$d^*(S(\epsilon), \mathbf{w}_t) = D - d_{\text{local}}(S(\epsilon), \mathbf{w}_t). \qquad (3.3)$$

Our theory provides several important insights on the nature of threshold training dimension. Firstly, small threshold training dimensions can only arise if the local angular dimension of the loss sublevel set $S(\epsilon)$ about the initialization $\mathbf{w}_t$ is close to the ambient dimension. Second, as $\epsilon$ increases, $S(\epsilon)$ becomes larger, with a larger $d_{\text{local}}(S(\epsilon), \mathbf{w}_t)$, and consequently a smaller threshold training dimension. Similarly, if $\mathbf{w}_t$ is closer to $S(\epsilon)$, then $d_{\text{local}}(S(\epsilon), \mathbf{w}_t)$ will be larger, and the threshold training dimension will also be lower (see fig. 4). This observation accounts for the observed decrease in threshold training dimension with increased burn-in time $t$. Presumably, burning in information into the initialization $\mathbf{w}_t$ for a longer time $t$ brings the initialization closer to the sublevel set $S(\epsilon)$, making it easier to hit with a random subspace of lower dimension. This effect is akin to staring out into the night sky in a single random direction and asking with what probability we will see the moon; this probability increases the closer we are to the moon.

### 3.2 A PARADIGMATIC LOSS LANDSCAPE EXAMPLE: THE QUADRATIC WELL

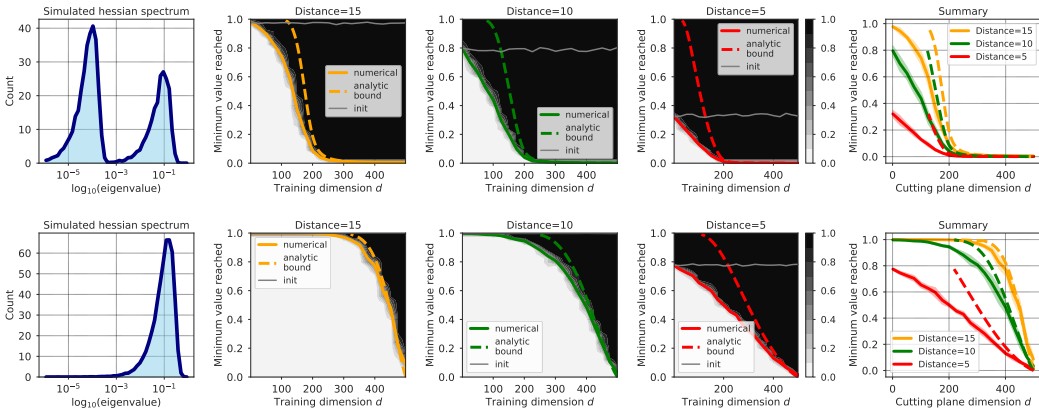

Figure 5: A comparison between simulated results and our analytic upper bound for threshold training dimension of sublevel sets on a synthetic quadratic well. The middle 3 columns show the success probability $P_s(d, \epsilon, R)$ as a function of $d$ and $\epsilon$ for three different values of the distance $R$ between initialization and the global minimum, clearly exhibiting a phase transition (black and white maps). This success probability is estimated from a numerical experiment across 10 runs and the estimated threshold training dimensions are shown as solid curves. Our analytic upper bounds on threshold training dimension obtained from our lower bound on local angular dimension in Eq. 3.4 are shown as dashed curves. The top row shows the case of a quadratic basin with a bimodal split of eigenvalues; the local angular dimension is approximately the number of long directions (small eigenvalues) and we start hitting low-loss sublevel sets at $D/2$ as expected. The bottom row shows the case of a continuous bulk spectrum. In both cases, threshold training dimension is lowered as the distance $R$ is decreased. The upper bound is tighter when $\epsilon$ is close to 0, the regime of we are most interested in.

To illustrate our theory, we work out the paradigmatic example of a quadratic loss function $\mathcal{L}(\mathbf{w}) = \frac{1}{2}\mathbf{w}^T\mathbf{H}\mathbf{w}$ where $\mathbf{w} \in \mathbb{R}^d$ and $\mathbf{H} \in \mathbb{R}^{D \times D}$ is a symmetric, positive definite Hessian matrix. A sublevel set $S(\epsilon)$ of the quadratic well is an ellipsoidal body with principal axes along the eigenvectors $\hat{\mathbf{e}}_i$ of $\mathbf{H}$. The radius $r_i$ along principal axis $\hat{\mathbf{e}}_i$ obeys $\frac{1}{2}\lambda_i r_i^2 = \epsilon$ where $\lambda_i$ is the eigenvalue. Thus $r_i = \sqrt{2\epsilon/\lambda_i}$, and so a large (small) Hessian eigenvalue leads to a narrow (wide) radius along each principal axis of the ellipsoid. The overall squared Gaussian width of the sublevel set obeys $w^2(S(\epsilon)) \sim 2\epsilon \, \text{Tr}(\mathbf{H}^{-1}) = \sum_i r_i^2$, where $\sim$ denotes bounded above and below by this expression times positive constants (Vershynin, 2018).

We next consider training within a random subspace of dimension $d$ starting from some initialization $\mathbf{w}_0$. To compute the probability the subspace hits the sublevel set $S(\epsilon)$, as illustrated in Fig. 4, we must project this ellipsoidal sublevel set onto the surface of the unit sphere centered at $\mathbf{w}_0$. The Gaussian width of this projection $\mathrm{proj}_{\mathbf{w}_0}(S(\epsilon))$ depends on the distance $R \equiv ||\mathbf{w}_0||$ from the initialization to the global minimum at $\mathbf{w} = \mathbf{0}$ (i.e. it should increase with decreasing $R$). We can develop a crude approximation to this width as follows. Assuming $D \gg 1$, the direction $\hat{\mathbf{e}}_i$ will be approximately orthogonal to $\mathbf{w}_0$, so that $|\hat{\mathbf{e}}_i \cdot \mathbf{x}_0| \ll R$. The distance between the tip of the ellipsoid at radius $r_i$ along principal axis $\mathbf{e}_i$ and the initialization $\mathbf{w}_0$ is therefore $\rho_i = \sqrt{R^2 + r_i^2}$. The ellipse's radius $r_i$ then gets scaled down to approximately $r_i / \sqrt{R^2 + r_i^2}$ when projected onto the surface of the unit sphere. Note the subtlety in this derivation is that the point actually projected onto the sphere is where a line through the center of the sphere lies tangent to the ellipse rather than the point of fullest extent. As a result, $r_i / \sqrt{R^2 + r_i^2}$ provides a lower bound to the projected extent on the circle. This is formalized in the appendix along with an explanation as to why this bound becomes looser with decreasing $R$. Taken together, a lower bound on the local angular dimension of $S(\epsilon)$ about $\mathbf{w}_0$ is:

$$d_{\mathrm{local}}(\epsilon, R) = w^2\big(\mathrm{proj}_{\mathbf{w}_0}(S(\epsilon))\big) \gtrsim \sum_i \frac{r_i^2}{R^2 + r_i^2}\,, \qquad (3.4)$$

where again $r_i = \sqrt{2\epsilon/\lambda_i}$. In Fig. 5, we plot the corresponding upper bound on the threshold training dimension, i.e. $D - d_{\mathrm{local}}(\epsilon, R)$ alongside simulated results for two different Hessian spectra.

## 4 CHARACTERIZING AND COMPARING THE SPACE OF PRUNING METHODS

Training within random subspaces is primarily a scientific tool to explore loss landscapes. It further has the advantage that we can explain theoretically why the number of degrees of freedom required to train can be far fewer than the number of parameters, as described above. However, there are many other pruning methods of interest. For example, the top row of Table 1 focuses on pruning to axis-aligned subspaces, starting from random weight pruning, to lottery tickets which use information from training to prune weights, and/or choose the initialization if not rewound to init. As one moves from left to right, one achieves better pruning (fewer degrees of freedom for a given accuracy). Our analysis can be viewed as relaxing the axis-aligned constraint to pruning to general subspaces (second row of Table 1), either not using training at all (random affine subspaces) or using information from training to only to choose the init (burn in affine subspaces). This analogy naturally leads to the notion of lottery subspaces described below (an analog of lottery tickets with axis-alignment relaxed to general subspaces) either rewound to init or not (last two entries of Table 1). We compare the methods we have theoretically analyzed (random and burn-in affine subspaces) to popular methods like lottery tickets rewound to init, and our new method of lottery subspaces, in an effort understand the differential efficacy of various choices like axis-alignment, initialization, and the use of full training information to prune. A full investigation of table 1 however is the subject of future work.

Table 1: Taxonomy of Pruning Methods.

| | Training not used | Training used for init. only | Training used for pruning only | Training used for init. and pruning |
|---|---|---|---|---|
| **Axis-aligned subspaces** | Random weight pruning | Random weight pruning at step $t$ | Lottery tickets, rewound to init. | Lottery tickets, rewound to step $t$ |
| **General subspaces** | Random affine subspaces | Burn-in affine at step $t$ | Lottery subspaces | Lottery subspaces at step $t$ |

**Lottery subspaces.** We first train the network in the full space starting from an initialization $\mathbf{w}_0$. We then form the matrix $\mathbf{U}_d \in \mathbb{R}^{D \times d}$ whose $d$ columns are the top $d$ principal components of entire the training trajectory $\mathbf{w}_{0:T}$ (see Appendix for details). We then train within the subspace $\mathbf{w}(\boldsymbol{\theta}) = \mathbf{U}_d \boldsymbol{\theta} + \mathbf{w}_t$ starting from a rewound initialization $\mathbf{w}_t$ ($t = 0$ is rewinding to the original init).

Since the subspace is optimized to match the top $d$ dimensions of the training trajectory, we expect lottery subspaces to achieve much higher accuracies for a given training dimension than random or potentially even burn-in affine subspaces. This expectation is indeed borne out in Fig. 3 (purple lines above all other lines). Intriguingly, *very* few lottery subspace training dimensions (in the range of 20 to 60 depending on the dataset and architecture) are required to attain full accuracy, and thus lottery

subspaces can set a (potentially optimistic) target for what accuracies might be attainable by practical network pruning methods as a function of training dimension.

Figure 6: Accuracy vs. compression ratio for the same data. Compression ratio is defined the number of parameters in the full model over the dimension of the subspace ($D/d$). The dimensions of the full parameter space for the experiments with CIFAR-10 are 25.6k for `Conv-2`, 66.5k for `Conv-3`, and 272.5k for `ResNet20`. On the other two datasets, the full parameter space is 20.5k for `Conv-2`, 61.5k for `Conv-3`, and 272.2k for `ResNet20`. The curve for each lottery ticket experiment summarizes data for at least 5 runs. For all other experiments, the curve summarizes data for 10 runs for `Conv-2`, 5 runs for `Conv-3`, and 3 runs for `ResNet20`. Black dotted lines are the accuracy of the full model run for the same number of epochs.

**Empirical comparison of pruning methods.** Figure 6 presents empirical results comparing a subset of the methods in table 1: random affine subspaces, burn-in affine subspaces, lottery subspaces, and lottery tickets plotted against model compression ratio (defined as parameters in full model over parameters, or training dimension, in restricted model). The lottery tickets were constructed by training for 2 epochs, performing magnitude pruning of weights and biases, rewinding to initialization, and then training for the same number of epochs as the other methods. Note that lottery tickets are created by pruning the full model (increasing compression ratio) in contrast to all other methods which are built up from a single dimension (decreasing compression ratio). We observe lottery subspaces significantly outperform random subspaces and lottery tickets at low training dimensions (high compression ratios), and we explore the spectrum of these spaces in more detail in the Appendix.

The comparison to lottery tickets at low compression ratios is limited by the fact that it is computationally expensive to project to higher dimensional subspaces and thus the highest training dimension we used was 4096. In the regions where the experiments overlap, the lottery tickets do not outperform random affine subspaces, indicating that they are not gaining an advantage from the training information they utilize. A notable exception is `Conv-2` on CIFAR-10 in which the lottery tickets do outperform random affine subspaces. Finally, we note lottery tickets do not perform well at high compression ratios due to the phenomenon of layer collapse, where an entire layer gets pruned.

## 5 CONCLUSION

The surprising ability of pruning methods like lottery tickets to achieve high accuracy with very few well-chosen parameters, and even higher accuracy if not rewound to initialization but to a later point in training, has garnered great interest in deep learning but has been hard to analyze. In this paper, we focused on gaining theoretical insight into when and why training within a random subspace starting at different initializations (or burn-ins) along a full training trajectory can achieve a given low loss $\epsilon$. We find that this can occur only when the local angular dimension of the loss sublevel set $S(\epsilon)$ about the initialization is high, or close to the ambient dimension $D$. Our theory also explains geometrically why longer burn-in lowers the number of degrees of freedom required to train to a given accuracy. This is analogous to how rewinding to a later point in training reduces the size of lottery tickets, and indeed may share a similar mechanism. Overall, these theoretical insights and comparisons begin to provide a high-dimensional geometric framework to understand and assess the efficacy of a wide range of network pruning methods at or beyond initialization.

ACKNOWLEDGEMENTS

B.W.L. was supported by the Department of Energy Computational Science Graduate Fellowship program (DE-FG02-97ER25308). S.G. thanks the Simons Foundation, NTT Research and an NSF Career award for funding while at Stanford.

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

## A EXPERIMENT SUPPLEMENT

The core experiment code is available on Github: `https://github.com/ganguli-lab/degrees-of-freedom`. The three top-level scripts are `burn_in_subspace.py`, `lottery_subspace.py`, and `lottery_ticket.py`. Random affine experiments were run by setting the parameter `init_iters` to 0 in the burn-in subspace code. The primary automatic differentiation framework used for the experiments was JAX Bradbury et al. (2018). The code was developed and tested using JAX v0.1.74, JAXlib v0.1.52, and Flax v0.2.0 and run on an internal cluster using NVIDIA TITAN Xp GPU's.

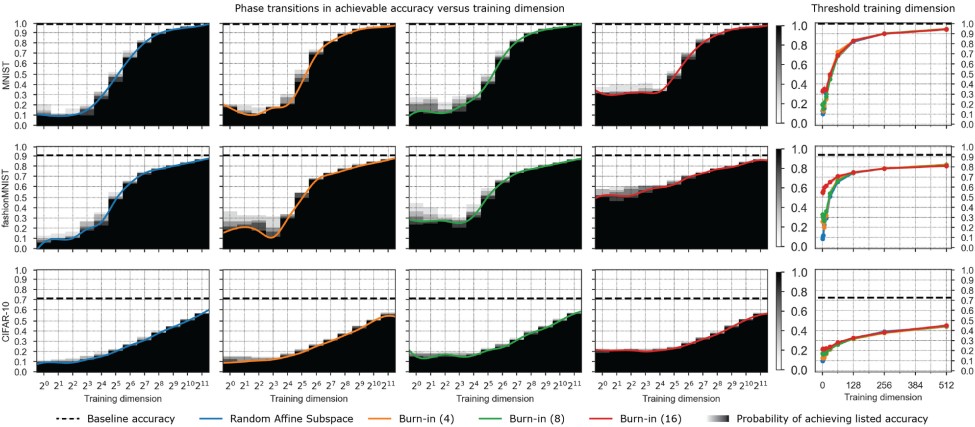

Figure 7: An empirical phase transition in training success on 3 datasets (3 rows) for a `Conv-3` comparing random affine subspaces (column 1) and burn-in affine subspaces with $t = 4, 8, 16$ burn-in steps (columns 2,3,4). The black-white color maps indicate the empirically measured success probability $P_s(d, \epsilon, t)$ in (2.3) in hitting a training accuracy super-level set. This success probability is estimated by training on 5 runs at every training dimension $d$. The horizontal dashed line represents the baseline accuracy obtained by training the full model for the same number of epochs. The colored curves indicate the threshold training dimension $d^*(\epsilon, t, \delta)$ in definition 2.1 for $\delta = 0.2$.

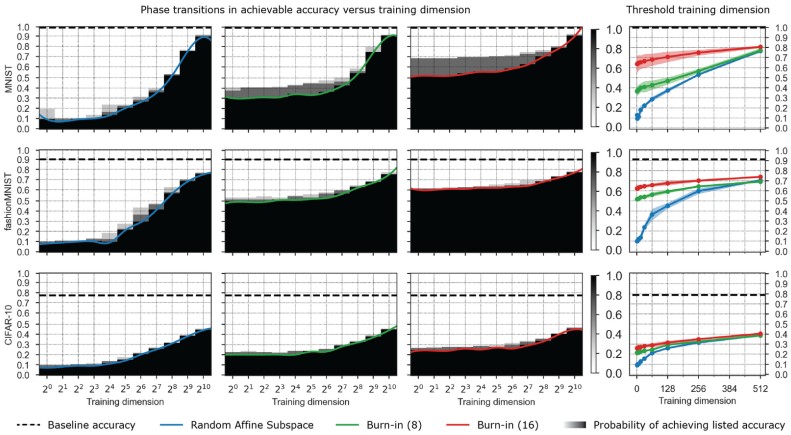

Figure 8: An empirical phase transition in training success on 3 datasets (3 rows) for a `ResNet20` comparing random affine subspaces (column 1) and burn-in affine subspaces with $t = 8, 16$ burn-in steps (columns 2,3). The black-white color maps indicate the empirically measured success probability $P_s(d, \epsilon, t)$ in (2.3) in hitting a training accuracy super-level set. This success probability is estimated by training on 3 runs at every training dimension $d$. The horizontal dashed line represents the baseline accuracy obtained by training the full model for the same number of epochs. The colored curves indicate the threshold training dimension $d^*(\epsilon, t, \delta)$ in definition 2.1 for $\delta = 0.33$.

Figures 7 and 8 show the corresponding empirical probability plots for the two other models considered in this paper: `Conv-3` and `ResNet20`. These plots are constructed in the same manner as fig. 2 except a larger value of $\delta$ was used since fewer runs were conducted ($\delta$ was always chosen such that all but one of the runs had to successfully hit a training accuracy super-level set). The data in these plots is from the same runs as figs. 3 and 6.

## A.1 COMPARISON TO LINEARIZED NETWORKS (NEURAL TANGENT KERNEL)

For general neural networks, we do not expect to be able bound the local angular dimension; instead, we use the relationship between the threshold training dimension and local angular dimension to empirically probe this important property of the loss landscape as in the experiments of fig. 3. For a single basin, we can consider the second-order approximation to the landscape at the optimum which yields a quadratic well based on the spectrum of the Hessian at this point, corresponding to the experiments presented in fig. 5 using a well with the appropriate spectrum. In this section, we consider how linearizing via the neural tangent kernel (NTK) can be used as a tool to better approximate the landscape of the network around a single basin while being potentially more amenable to theoretical characterization than the full network.

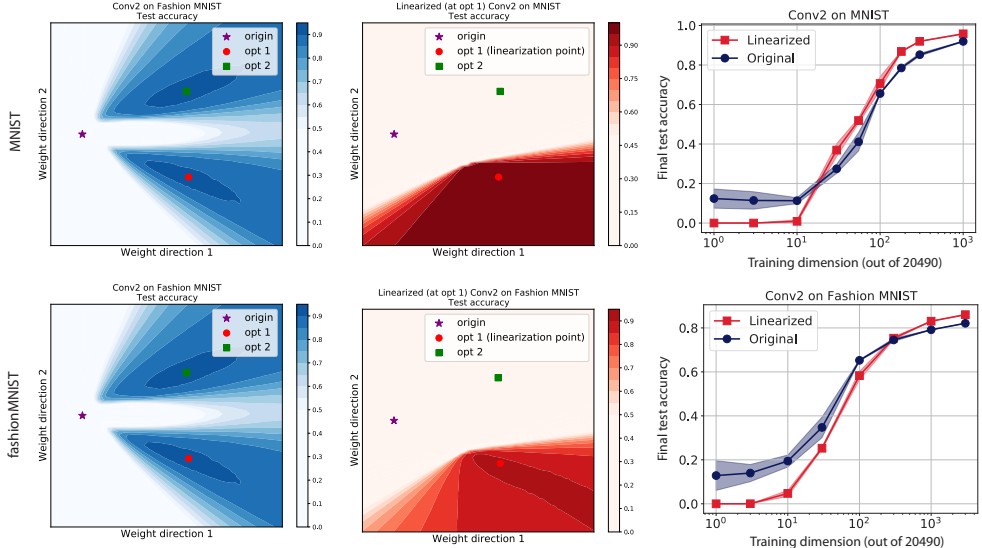

Figure 9: Empirical comparison of the threshold training dimension for the full landscape vs. the linearized model around a specific local optimum. These experiments were run on the `Conv-2` model; the top row is on MNIST and the bottom is on Fashion MNIST. The column show a 2-dimensional cut of the test loss landscape defined by the initialization point and two optimum found via training. The second column shows the same cut but for the linearized model around optimum 1. Finally, the right column shows the threshold training dimension for both the original and linearized model.

For this experiment we first train in the full network starting from initialization $\mathbf{w}_0 \in \mathbb{R}^D$ until we find a local optimum $\mathbf{w}_{\text{opt}}$. Instead of using the second-order approximation around this optimum given by the Hessian, we linearize the network around $\mathbf{w}_{\text{opt}}$ via the NTK (Jacot et al., 2018). In essence, if $f(\mathbf{w}, \mathbf{x})$ is the function that outputs the $i$th logit for a given input $\mathbf{x}$ we instead consider the following approximation which is a linear function in $\mathbf{w}$:

$$f(\mathbf{w}_{\text{opt}} + \mathbf{w}, \mathbf{x}) \approx f(\mathbf{w}_{\text{opt}}, \mathbf{x}) + [\nabla_{\mathbf{w}} f(\mathbf{w}_{\text{opt}}, \mathbf{x})]^{\mathsf{T}} \mathbf{w} := A(\mathbf{w}_{\text{opt}}, \mathbf{x}) + \mathbf{B}(\mathbf{w}_{\text{opt}}, \mathbf{x}) \cdot \mathbf{w}$$

At $\mathbf{w}_{\text{opt}}$, the full and linearized network are identical; however, in the linearized network there is only one basin which is around $\mathbf{w}_{\text{opt}}$. We then compare these networks by returning to the initialization point $\mathbf{w}_0$ and perform the experiment training within random affine subspaces across a range of dimensions in both the full and linearized network.

Figure 9 shows the results of this experiment for both MNIST and Fashion MNIST using the model `Conv-2`. In these two settings, the threshold training dimension of the linearized model approximates this property of the full model fairly well, indicating promise as a useful approximation to the true loss landscape around a basin. Thus, we consider theoretically characterizing the local angular dimension of these linearized models interesting future work.

## A.2 SPECTRA OF LOTTERY SUBSPACES

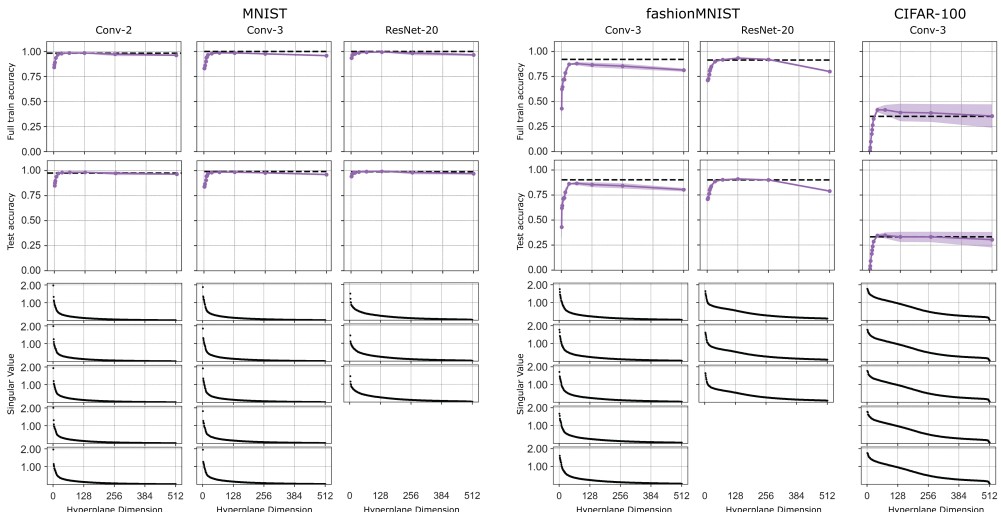

Figure 10: In the main text, we plot a running max accuracy applied to the individual runs because the subspaces are nested and we are only concerned with the existence of an intersection. The accuracies are plotted here without this preprocessing step for comparison with the spectra. **Left:** Singular values for the lottery subspace experiments on MNIST. **Right:** Singular values for the lottery subspace experiments on Fashion MNIST and an additional run on CIFAR-100. Only the first 5 spectra (out of 10) are shown for `Conv-2`. Directions were added in order of descending singular values.

In our experiments, we formed lottery subspaces by storing the directions traveled during a full training trajectory and then finding the singular value decomposition of this matrix. As we increased the subspace dimension, directions were added in order of descending singular values. Figure 10 and the left panel of fig. 11 show the associated spectra for the results presented in figs. 3 and 6. Note that in the main text figures we plot the accuracies as a running max over the current and smaller dimensions. This is because the subspaces are nested such that if we increase the dimension and find a point of lower accuracy, it indicates a failure of optimization to find the intersection as the higher accuracy point is still in the subspace. In these supplement figures, we plot the recorded accuracies without this processing step for completeness. We see that in several cases this optimization failure did occur as we moved to higher dimensions; we suspect this is related to how quickly the singular values fall off meaning the higher dimensions we add are much less informative.

The spectra are aligned with the train and test accuracy plots such that the value directly below a point on the curve corresponds to the singular value of the last dimension added to the subspace. There were 10 runs for `Conv-2`, 5 for `Conv-3`, and 3 for `ResNet20`. Only the first 5 out of 10 runs are displayed for the experiments with `Conv-2`. No significant deviations were observed in the remaining runs.

From these plots, we observe that the spectra for a given dataset are generally consistent across architectures. In addition, the decrease in accuracy after a certain dimension (particularly for CIFAR-10) corresponds to the singular values of the added dimensions falling off towards 0.

The right panel of fig. 11 shows a tangential observation that lottery subspaces for CIFAR-10 display a sharp transition in accuracy at $d = 10$. This provides additions evidence for the conjecture explored by Gur-Ari et al. (2018), Fort & Ganguli (2019), and Papyan (2020) that the sharpest directions of the

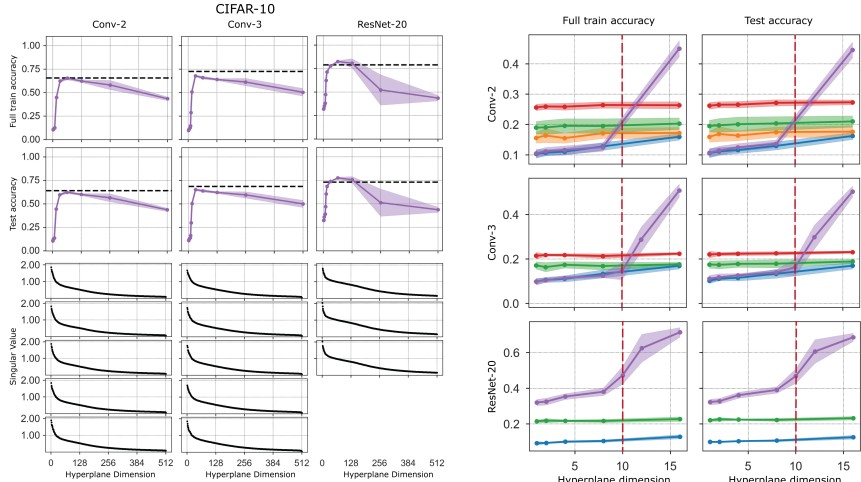

Figure 11: In the main text, we plot a running max accuracy applied to the individual runs beacuse the subspaces are nested and we are only concerned with the existence of an intersection. The accuracies are plotted here without this preprocessing step for comparison with the spectra. **Left:** Singular values for the lottery subspace experiments on CIFAR-10. Only the first 5 spectra (out of 10) are shown for `Conv-2`. Directions were added in order of descending singular values. **Right:** Lottery subspaces display accuracy transition around $d = 10$ for the dataset CIFAR-10. Provides additional evidence for the conjecture that the shaprest directions of the Hessian are each associated with a class but no learning happens in them.

Hessian and the most prominent logit gradients are each associated with a class. Very little learning happens in these directions, but during optimization you bounce up and down along them so that the are prominent in the SVD of the gradients. This predicts exactly the behavior observed.

## A.3 ACCURACY OF BURN-IN INITIALIZATION

Figure 12 shows a subset of the random affine and burn-in affine subspace experiments with a value plotted at dimension 0 to indicate the accuracy of the random or burn-in initialization. This is to give context for what sublevel set the burn-in methods are starting out, enabling us to evaluate whether they are indeed reducing the threshold training dimension of sublevel sets with higher accuracy. In most cases, as we increase dimension the burn-in experiments increase in accuracy above their initialization and at a faster pace than the random affine subspaces. A notable exception is `Conv-3` on MNIST in which the burn-in methods appear to provide no advantage.

## A.4 HYPERPARAMETERS

Random hyperplanes were chosen by sampling a $D \times d$ matrix of independent, standard Gaussians and then normalizing the columns to 1. This is equivalent to sampling uniformly from the Grassmanian as required by theorem 3.2. Optimization restricted to an affine subspace was done using `Adam` Kingma & Ba (2014) with $\beta_1 = 0.9$, $\beta_2 = 0.999$, and $\epsilon = 10^{-7}$. We explored using $5 \cdot 10^{-2}$ and $10^{-2}$ for the learning rate but $5 \cdot 10^{-2}$ worked substantially better for this restricted optimization and was used in all experiments; a batch size of 128 was used. The full model runs used the better result of $5 \cdot 10^{-2}$ and $10^{-2}$ for the learning rate. `ResNet20v1` was run with on-the-fly batch normalization Ioffe & Szegedy (2015), meaning we simply use the mean and variance of the current batch rather than maintaining a running average. Table 2 shows the number of epochs used for each dataset and architecture combination across all experiments. 3 epochs was chosen by default and then increased if the full model was not close to convergence.

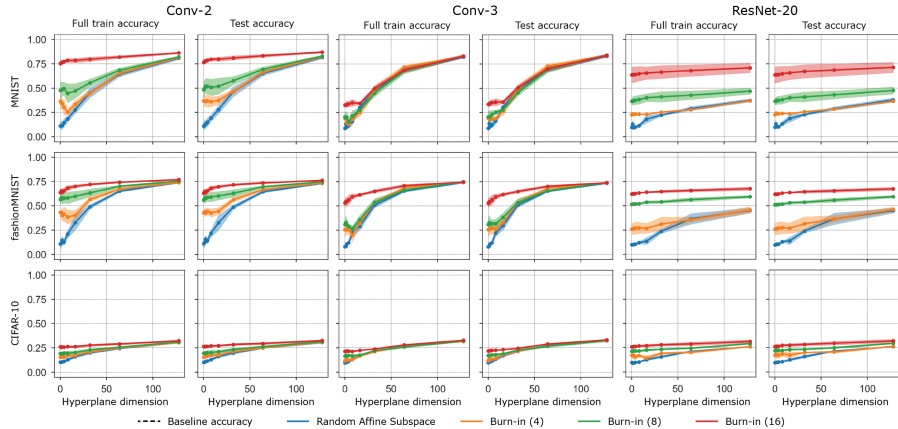

Figure 12: First 128 dimensions for a subset of the random affine and burn-in affine subspace experiments. The plots include a value at dimension 0 which indicates the accuracy of the random initialization or the burn-in initialization.

Table 2: Epochs Used for Experiments.

| Dataset | MNIST | Fashion MNIST | CIFAR-10 | CIFAR-100 | SVHN |
|---|---|---|---|---|---|
| Conv-2 | 3 epochs | 3 epochs | 4 epochs | - | 4 epochs |
| Conv-3 | 4 epochs | 5 epochs | 5 epochs | 5 epochs | - |
| ResNet20 | 3 epochs | 3 epochs | 4 epochs | - | - |

## B    THEORY SUPPLEMENT

In this section, we provide additional details for our study of the threshold training dimension of the sublevel sets of quadratic wells. We also derive the threshold training dimension of affine subspaces to provide further intuition.

### B.1    PROOF: GAUSSIAN WIDTH OF SUBLEVEL SETS OF THE QUADRATIC WELL

In our derivation of eq. (3.4), we employ the result that the Gaussian width squared of quadratic well sublevel sets is bounded as $w^2(S(\epsilon)) \sim 2\epsilon \operatorname{Tr}(\mathbf{H}^{-1}) = \sum_i r_i^2$, i.e. bounded above and below by this expression times positive constants. This follows from well-established bounds on the Gaussian width of an ellipsoid which we now prove.

In our proof, we will use an equivalent expression for the Gaussian width of set $S$:

$$w(S) := \frac{1}{2} \mathbb{E} \sup_{\mathbf{x}, \mathbf{y} \in S} \langle \mathbf{g}, \mathbf{x} - \mathbf{y} \rangle = \mathbb{E} \sup_{\mathbf{x} \in S} \langle \mathbf{g}, \mathbf{x} \rangle, \quad \mathbf{g} \sim \mathcal{N}(\mathbf{0}, \mathbf{I}_{D \times D}).$$

**Lemma B.1** (Gaussian width of ellipsoid). *Let $\mathcal{E}$ be an ellipsoid in $\mathbb{R}^D$ defined by the vector $\mathbf{r} \in \mathbb{R}^D$ with strictly positive entries as:*

$$\mathcal{E} := \left\{ \mathbf{x} \in \mathbb{R}^D \, \middle| \, \sum_{j=1}^{D} \frac{x_j^2}{r_j^2} \le 1 \right\}$$

*Then $w(\mathcal{E})^2$ or the Gaussian width squared of the ellipsoid satisfies the following bounds:*

$$\sqrt{\frac{2}{\pi}} \sum_{j=1}^{D} r_j^2 \le w(\mathcal{E})^2 \le \sum_{j=1}^{D} r_j^2$$

*Proof.* Let $\mathbf{g} \sim \mathcal{N}(\mathbf{0}, \mathbf{I}_{D \times D})$. Then we upper-bound $w(\mathcal{E})$ by the following steps:

$$
\begin{aligned}
w(\mathcal{E}) &= \mathbb{E}_{\mathbf{g}} \left[ \sup_{\mathbf{x} \in \mathcal{E}} \sum_{i=1}^{D} g_i x_i \right] \\
&= \mathbb{E}_{\mathbf{g}} \left[ \sup_{\mathbf{x} \in \mathcal{E}} \sum_{i=1}^{D} \frac{x_i}{r_i} g_i r_i \right] && \frac{r_i}{r_i} = 1 \\
&\le \mathbb{E}_{\mathbf{g}} \left[ \sup_{\mathbf{x} \in \mathcal{E}} \left( \sum_{i=1}^{D} \frac{x_i^2}{r_i^2} \right)^{1/2} \left( \sum_{i=1}^{D} g_i^2 r_i^2 \right)^{1/2} \right] && \text{Cauchy-Schwarz inequality} \\
&\le \mathbb{E}_{\mathbf{g}} \left[ \left( \sum_{i=1}^{D} g_i^2 r_i^2 \right)^{1/2} \right] && \text{Definition of } \mathcal{E} \\
&\le \sqrt{ \mathbb{E}_{\mathbf{g}} \left[ \sum_{i=1}^{D} g_i^2 r_i^2 \right]} && \text{Jensen's inequality} \\
&\le \left( \sum_{i=1}^{D} r_i^2 \right)^{1/2} && \mathbb{E}[w_i^2] = 1
\end{aligned}
$$

giving the upper bound in the lemma. For the lower bound, we will begin with a general lower bound for Gaussian widths using two facts. The first is that if $\epsilon_i$ are i.i.d. Rademacher random variables and, then $\epsilon_i |g_i| \sim \mathcal{N}(0, 1)$. Second, we have:

$$\mathbb{E}[|g_i|] = \frac{1}{2\pi} \int_{-\infty}^{\infty} |y| e^{-y^2/2} dy = \frac{2}{\sqrt{2\pi}} \int_0^{\infty} y e^{-y^2/2} = \frac{2}{\pi}$$

Then for the Gaussian width of a general set:

$$
\begin{aligned}
w(S) &= \mathbb{E} \left[ \sup_{x \in S} \sum_{i=1}^{D} w_i x_i \right] \\
&= \mathbb{E}_{\epsilon} \left[ \mathbb{E}_w \left[ \sup_{x \in S} \sum_{i=1}^{n} \epsilon_i |g_i| \cdot x_i \, \middle| \, \epsilon_{1:n} \right] \right] && \text{Using } \epsilon_i |g_i| \sim \mathcal{N}(0, 1) \\
&\ge \mathbb{E}_{\epsilon} \left[ \sup_{x \in S} \sum_{i=1}^{D} \epsilon_i \mathbb{E}[|g_i|] x_i \right] && \text{Jensen's Inequality} \\
&= \sqrt{\frac{2}{\pi}} \mathbb{E} \left[ \sup_{x \in S} \sum_{i=1}^{D} \epsilon_i x_i \right]
\end{aligned}
$$

All that remains for our lower bound is to show that for the ellipsoid $\mathbb{E}\left[\sup_{x \in \mathcal{E}} \sum_{i=1}^{D} \epsilon_i x_i\right] = \left(\sum_{i=1}^{D} r_i^2\right)^{1/2}$. We begin by showing it is an upper-bound:

$$
\begin{aligned}
\mathbb{E}\left[\sup_{x \in \mathcal{E}} \sum_{i=1}^{D} \epsilon_i x_i\right] &= \sup_{x \in \mathcal{E}} \sum_{i=1}^{D} |x_i| && \text{Using } \mathcal{E} \text{ is symmetric} \\
&= \sup_{x \in \mathcal{E}} \sum_{i=1}^{D} \left|\frac{x_i r_i}{r_i}\right| && \frac{r_i}{r_i} = 1 \\
&\leq \sup_{\mathbf{x} \in \mathcal{E}} \left(\sum_{i=1}^{D} \frac{x_i^2}{r_i^2}\right)^{1/2} \left(\sum_{i=1}^{D} r_i^2\right)^{1/2} && \text{Cauchy-Schwarz inequality} \\
&= \left(\sum_{i=1}^{D} r_i^2\right)^{1/2} && \text{Definition of } \mathcal{E}
\end{aligned}
$$

In the first line, we mean that $\mathcal{E}$ is symmetric about the origin such that we can use $\epsilon_i = 1$ for all $i$ without loss of generality. Finally, consider $\mathbf{x}$ such that $x_i = r_i^2 / \left(\sum_{i=1}^{D} r_i^2\right)^{1/2}$. For this choice we have $\mathbf{x} \in \mathcal{E}$ and:

$$
\sum_{i=1}^{D} |x_i| = \sum_{i=1}^{D} \frac{r_i^2}{\left(\sum_{i=1}^{D} r_i^2\right)^{1/2}} = \left(\sum_{i=1}^{D} r_i^2\right)^{1/2}
$$

showing that equality is obtained in the bound. Putting these steps together yields the overall desired lower bound:

$$
w(\mathcal{E}) \geq \sqrt{\frac{2}{\pi}} \cdot \mathbb{E}\left[\sup_{x \in \mathcal{E}} \sum_{i=1}^{D} \epsilon_i x_i\right] = \sqrt{\frac{2}{\pi}} \cdot \left(\sum_{i=1}^{D} r_i^2\right)^{1/2}
$$

$\square$

With this bound in hand, we can immediately obtain the following corollary for a quadratic well defined by Hessian $\mathbf{H}$. The Gaussian width is invariant under affine transformation so we can shift the well to the origin. Then note that $S(\epsilon)$ is an ellipsoid with $r_i = \sqrt{2\epsilon/\lambda_i}$ and thus $\sum_i r_i^2 = \epsilon \operatorname{Tr}(\mathbf{H}^{-1})$.

**Corollary B.1** (Gaussian width of quadratic sublevel sets). *Consider a quadratic well defined by Hessian $\mathbf{H} \in \mathbb{R}^{D \times D}$. Then the Gaussian width squared of the associated sublevel sets $S(\epsilon)$ obey the following bound:*

$$
\sqrt{\frac{2}{\pi}} \cdot 2\epsilon \operatorname{Tr}(\mathbf{H}^{-1}) \leq w^2(S(\epsilon)) \leq 2\epsilon \operatorname{Tr}(\mathbf{H}^{-1})
$$

### B.2 DETAILS ON THRESHOLD TRAINING DIMENSION UPPER BOUND

In section 3.2, we consider the projection of ellipsoidal sublevel sets onto the surface of a unit sphere centered at $\mathbf{w_0}$. The Gaussian width of this projection $\operatorname{proj}_{\mathbf{w_0}}(S(\epsilon))$ will depend on the distance $R \equiv \|\mathbf{w_0}\|$ from the initialization to the global minimum at $\mathbf{w} = \mathbf{0}$ (i.e. it should increase with decreasing $R$). We used a crude approximation to this width as follows. Assuming $D \gg 1$, the direction $\hat{\mathbf{e}}_i$ will be approximately orthogonal to $\mathbf{w_0}$, so that $|\hat{\mathbf{e}}_i \cdot \mathbf{x_0}| \ll R$. The distance between the tip of the ellipsoid at radius $r_i$ along principal axis $\mathbf{e}_i$ and the initialization $\mathbf{w_0}$ is therefore $\rho_i = \sqrt{R^2 + r_i^2}$. The ellipse's radius $r_i$ then gets scaled down to approximately $r_i / \sqrt{R^2 + r_i^2}$ when projected onto the surface of the unit sphere.

We now explain why this projected size is always a lower bound by illustrating the setup in two dimensions in fig. 13. As shown, the linear extent of the projection will always result from a line that is tangent to the ellipse. For an ellipse $(x/a)^2 + ((y - R)/b)^2 = 1$ and a line $y = cx$ in a two-dimensional space (we set the origin at the center of the unit circle), a line tangent to the ellipse

must satisfy $c = a/\sqrt{R^2 - b^2}$. That means that the linear extent of the projection on unit circle will be $a/\sqrt{a^2 + R^2 - b^2}$. For $a = 2\epsilon/\lambda_i$ and $R = R$, this is exactly Eq. 3.4 provided $b = 0$. The $b \neq 0$ will *always* make the linear projections *larger*, and therefore Eq. 3.4 will be a lower bound on the projected Gaussian width. Furthermore, this bound will be looser with decreasing $R$. We then obtain a corresponding upper bound on the threshold training dimension, i.e. $D - d_{\text{local}}(\epsilon, R)$.

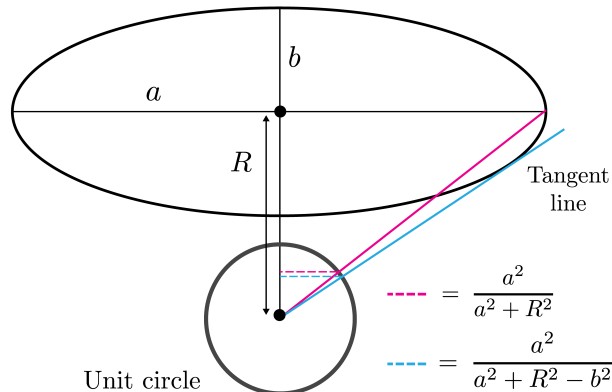

Figure 13: Illustration in two dimensions why the projection of the principal axes of an ellipse onto the unit circle will lower bound the size of the projected set. The linear extent of the projection will result from a line that lies tangent to the ellipse.

### B.3 THRESHOLD TRAINING DIMENSION OF AFFINE SUBSPACES

In Section 3.2, we considered the threshold training dimension of the sublevel sets of a quadratic well and showed that it depends on the distance from the initialization to the set, formalized in eq. (3.4). As a point of contrast, we include a derivation of the threshold training dimension of a random affine subspace in ambient dimension $D$ and demonstrate that this dimension does not depend on distance to the subspace. Intuitively this is because any dimension in the subspace is of infinite or zero extent, unlike the quadratic sublevel sets which have dimensions of finite extent.

Let us consider a $D$-dimensional space for which we have a randomly chosen $d$-dimensional affine subspace $A$ defined by a vector offset $\mathbf{x}_0 \in \mathbb{R}^D$ and a set of $d$ orthonormal basis vectors $\{\hat{\mathbf{v}}_i\}_{i=1}^d$ that we encapsulate into a matrix $\mathbf{M} \in \mathbb{R}^{d \times D}$. Let us consider another random $n$-dimensional affine subspace $B$. Our task is to find a point $\mathbf{x}^* \in A$ that has the minimum $\ell_2$ distance to the subspace $B$, i.e.:

$$\mathbf{x}^* = \text{argmin}_{\mathbf{x} \in A} \big\| \mathbf{x} - \text{argmin}_{\mathbf{x}' \in B} \|\mathbf{x} - \mathbf{x}'\|_2 \big\|_2$$

In words, we are looking for a point in the $d$-dimensional subspace $A$ that is as close as possible to its closest point in the $n$-dimensional subspace $B$. Furthermore, points within the subspace $A$ can be parametrized by a $d$-dimensional vector $\boldsymbol{\theta} \in \mathbb{R}^d$ as $\mathbf{x}(\boldsymbol{\theta}) = \boldsymbol{\theta} \mathbf{M} + \mathbf{x}_0 \in A$; for all choices of $\boldsymbol{\theta}$, the associated vector $\mathbf{x}$ is in the subspace $A$.

Without loss of generality, let us consider the case where the $n$ basis vectors of the subspace $B$ are aligned with the dimensions $D - n, D - n + 1, \ldots, D$ of the coordinate system (we can rotate our coordinate system such that this is true). Call the remaining axes $s = D - n$ the *short* directions of the subspace $B$. A distance from a point $\mathbf{x}$ to the subspace $B$ now depends only on its coordinates $1, 2, \ldots, s$. Under our assumption of the alignment of subspace $B$ we then have:

$$l^2(\mathbf{x}, B) := \text{argmin}_{\mathbf{x}' \in B} \|\mathbf{x} - \mathbf{x}'\|_2^2 = \sum_{i=1}^s x_i^2$$

The only coordinates influencing the distance are the first $s$ values, and thus let us consider a $\mathbb{R}^s$ subspace of the original $\mathbb{R}^D$ only including those without loss of generality. Now $\boldsymbol{\theta} \in \mathbb{R}^d$, $\mathbf{M} \in \mathbb{R}^{d \times s}$

and $\mathbf{x}_0 \in \mathbb{R}^d$, and the distance between a point within the subspace $A$ parameterized by the vector $\boldsymbol{\theta}$ is given by:

$$l^2\big(\mathbf{x}(\boldsymbol{\theta}), B\big) = \|\boldsymbol{\theta}\mathbf{M} + \mathbf{x}_0\|^2 .$$

The distance $l$ attains its minimum for

$$\partial_{\boldsymbol{\theta}} l^2\big(\mathbf{x}(\boldsymbol{\theta}), B\big) = 2 \cdot (\boldsymbol{\theta}\mathbf{M} + \mathbf{x}_0) \, \mathbf{M}^T = \mathbf{0}$$

yielding the optimality condition $\boldsymbol{\theta}^*\mathbf{M} = -\mathbf{x_0}$. There are 3 cases based on the relationship between $d$ and $s$.

**1. The overdetermined case,** $d > s$. In case $d > s = D - n$, the optimal $\boldsymbol{\theta}^* = -\mathbf{x}_0\mathbf{M}^{-1}$ belongs to a ($d - s = d + n - D$)-dimensional family of solutions that attain 0 distance to the plane $B$. In this case the affine subspaces $A$ and $B$ intersect and share a ($d + n - D$)-dimensional intersection.

**2. A unique solution case,** $d = s$. In case of $d = s = D - n$, the solution is a unique $\boldsymbol{\theta}^* = -\mathbf{x}_0\mathbf{M}^{-1}$. After plugging this back to the distance equation, we obtain $\boldsymbol{\theta}$ is

$$l^2(\mathbf{x}(\boldsymbol{\theta}^*), B) = \left\|-\mathbf{x}_0\mathbf{M}^{-1}\mathbf{M} + \mathbf{x}_0\right\|^2$$
$$= \|-\mathbf{x}_0 + \mathbf{x}_0\|^2 = 0.$$

The matrix $\mathbf{M}$ is square in this case and cancels out with its inverse $\mathbf{M}^{-1}$.

**3. An underdetermined case,** $d < s$. In case of $d < s$, there is generically no intersection between the subspaces. The inverse of $\mathbf{M}$ is now the Moore-Penrose inverse $\mathbf{M}^+$. Therefore the closest distance $\boldsymbol{\theta}$:

$$l^2\big(\mathbf{x}(\boldsymbol{\theta}^*), B\big) = \left\|-\mathbf{x}_0\mathbf{M}^+\mathbf{M} + \mathbf{x}_0\right\|^2$$

Before our restriction from $D \to s$ dimensions, the matrix $\mathbf{M}$ consisted of $d$ $D$-dimensional, mutually orthogonal vectors of unit length each. We will consider these vectors to be component-wise random, each component with variance $1/\sqrt{D}$ to satisfy this condition on average. After restricting our space to $s$ dimensions, $\mathbf{M}$'s vectors are reduced to $s$ components each, keeping their variance $1/\sqrt{D}$. They are still mutually orthogonal in expectation, but their length are reduced to $\sqrt{s}/\sqrt{D}$. The transpose of the inverse $\mathbf{M}^+$ consists of vectors of the same directions, with their lengths scaled up to $\sqrt{D}/\sqrt{s}$. That means that in expectation, $\mathbf{M}\mathbf{M}^+$ is a diagonal matrix with $d$ diagonal components set to 1, and the remainder being 0. The matrix $(\mathbf{I} - \mathbf{M}^+\mathbf{M})$ contains $(s - d)$ ones on its diagonal. The projection $\|\mathbf{x}_0(\mathbf{I} - \mathbf{M}^+\mathbf{M})\|^2$ is therefore of the expected value of $\|\mathbf{x}_0\|^2(s - d)^2/D$. The expected distance between the $d$-dimensional subspace $A$ and the $d$-dimensional subspace $B$ is:

$$\mathbb{E}\big[d(A, B)\big] \propto \begin{cases} \frac{\sqrt{D-n-d}}{\sqrt{D}} & n + d < D \,, \\ 0 & n + d \geq D \,. \end{cases} \qquad \square$$

To summarize, for a space of dimension $D$, two affine subspaces generically intersect provided that their dimensions $d_A$ and $d_B$ add up to at least the ambient (full) dimension of the space. The exact condition for intersection is $d_A + d_B \geq D$, and the threshold training dimension of subspace $B$ is $D - d$. This result provides two main points of contrast to the quadratic well:

- **Even extended directions are not infinite for the quadratic well.** While in the case of the affine subspaces even a slight non-coplanarity of the target affine subspace and the random training subspace will eventually lead to an intersection, this is not the case for the sublevel sets of the quadratic well. Even its small eigenvalues, i.e. shallow directions, will still have a finite extent for all finite $\epsilon$.

- **Distance independence of the threshold training dimension.** As a result of the dimensions having finite extent, the distance independence of threshold training dimension for affine subspaces does not carry over to the case of quadratic wells. In the main text, this dependence on distance is calculated by projecting the set onto the unit sphere around the initialization enabling us to apply Gordon's Escape Theorem.

