# OpenReview forum: "How many degrees of freedom do we need to train deep networks: a loss landscape perspective"
_ICLR.cc/2022/Conference — ICLR 2022 Poster_

### Official Review · Reviewer_L6aD · 2021-11-02

**Correctness:** 4
**Technical Novelty And Significance:** 3
**Empirical Novelty And Significance:** 3
**Recommendation:** 6
**Confidence:** 4

**Main Review:**

The article is well written and clear. It studies an interesting phenomenon with mix of empirical and theoretical results.

The empirical experiments are detailed and illustrate clearly the transition they are studying.

The theoretical results are nice and explained well. It is not much more than a corollary of Gordon's escape theorem, but I know of no previous work using it in this context.

The example analysis for quadratic loss is important, it gives an idea of how one could compute the local angular dimension in practice. I think the authors have missed an opportunity here to draw a link with the loss of DNNs. With the MSE loss and in the NTK regime (with a specific initialization and sufficiently many neurons in the hidden layers), it has been shown that the loss landscape around the training path is approximately quadratic (https://openreview.net/pdf?id=SkgscaNYPS), with spectrum equal to the spectrum of the NTK Gram matrix. In this setting, the local angular dimension of the sublevel sets of the DNN loss should be well approximated by the local angular dimension of the quadratic approximation. This would make it possible to compare the empirical transition with your theory in the DNN case directly.

**Summary Of The Paper:**

The paper studies DNNs training when restricted to a random affine subspace (centered either at the initialization point $x_0$ or at a point $x_t$ obtained by training an unrestricted network for $t$ steps, which they call burn-in), and the probability that a loss lower than $\epsilon$ can be reached with a subspace of dimension $d$. They observe a sharp transition between pairs $(\epsilon,d)$ where this probability is either almost $0$ or $1$. They observe that the lower the dimension, the less likely one is to reach a loss $\epsilon$.

The authors show that this transition can be understood in terms of the Gaussian width of the projection of the sublevel set $S(\epsilon)$ to the unit ball around the center point $x_0$ (or $x_t$), which they call the local angular dimension. Gordon's escape theorem makes it possible to bound the transition between the two phases as a function of the local angular dimension.

Calculating the local angular dimension of the sublevel sets of the loss of DNNs is difficult, instead the authors study a quadratic loss, giving an approximation for the local angular dimension in this case. They then compare the empirical transition and the bound on the transition in terms of the local angular dimension between the two phases and find a good agreement with the theory.

Finally they propose a notion of lottery subspaces inspired by the lottery tickets paper and compare their results.

**Summary Of The Review:**

The article is clear and well written. It studies an interesting phenomenon with mix of empirical and theoretical results. The theoretical results are clear and interesting though they are direct consequence of Gordon's escape theorem.

---

> ### Author Response · Authors · 2021-11-20
> **Response to Reviewer L6aD**
>
> **The example analysis for quadratic loss is important, it gives an idea of how one could compute the local angular dimension in practice. I think the authors have missed an opportunity here to draw a link with the loss of DNNs. With the MSE loss and in the NTK regime (with a specific initialization and sufficiently many neurons in the hidden layers), it has been shown that the loss landscape around the training path is approximately quadratic (https://openreview.net/pdf?id=SkgscaNYPS), with spectrum equal to the spectrum of the NTK Gram matrix. In this setting, the local angular dimension of the sublevel sets of the DNN loss should be well approximated by the local angular dimension of the quadratic approximation. This would make it possible to compare the empirical transition with your theory in the DNN case directly.**
>
> We think the reviewer's suggestion is an excellent one; in fact, we had recently begun to consider how to use networks linearized via the NTK as a reasonable approximation to the landscape that could remain theoretically tractable. In particular, we want to use the linearzation of the network around an optimum $\mathbf{w}_{\text{opt}}$ to approximate the true landscape in the neighborhood of this basin. While we were not able to fully complete the suggested comparison during the time allowed for discussion, we have updated the paper with new experiments that begin to push in this direction.
>
> In the revised manuscript, we have added a section to the supplement (Section A.1) that considers comparisons to networks linearized around a particular $\mathbf{w}_{\text{opt}}$ in the loss landscape via the NTK.  We show that for Conv-2 trained on MNIST and Fashion MNIST, the threshold training dimension of the linearized model approximates the same property of the full model fairly well for a nearby initialization point.  We think this similarity points to theoretically characterizing this linearized model (which has a single basin rather than the much more complicated "web" of solutions in full networks) as a promising direction.  These experiments were done with the cross-entropy loss to match the work in the main paper; we agree with the reviewer that the MSE loss would be a better place to start for theory.

---

### Official Review · Reviewer_fVom · 2021-11-02

**Correctness:** 4
**Technical Novelty And Significance:** 4
**Empirical Novelty And Significance:** 3
**Recommendation:** 6
**Confidence:** 4

**Main Review:**

Strengths:

Theory:
This paper introduces a novel concept using the Gordon escape theorem to understand the effect of training in lower-dimensional random subspace of the NN weight space, especially after a burn-in period. Specifically the theorem shows the existence of a phase transition in the success probability of training and this is supported by the experiments as well.

Experiments:
They also provide a new training/pruning method based on this phenomena, which performs better than other existing training methods, especially in low-training dimensions

Weaknesses:

The theorem is a first such characterization, but it is not clear as to the difficulty of estimating the threshold dimensions for more than quadratic loss functions, potentially when involving deep ReLU networks for instance. It would be nice to have a discussion of the challenges for at least shallow networks (1 hidden layer ReLU network for example).

Another clarification is in Eq 2.2, the randomness of A is from a Gaussian distribution. In the statement of Gordon escape theorem it would be useful to clarify what "uniformly from a Grassmanian mean"? Are there any restrictions on how the random subspace is generated?
As a continuation, in the Lottery subspaces algorithm, the matrix U is generated by the dynamics of GD on the loss function. Does the Gordon escape theorem still apply for this random process?

Is it possible to infer in the pruning algorithm and the Gordon escape theorem, how the trainable network depends on depth or width of the original network? It would be interesting to see at least experimentally the change in depth/width for large classes of problems, as this ties in with numerous results on expressivity that say in general , increasing depth is more useful than increasing width (from the lens of expressivity alone!).

It would be useful if the authors can clarify the dependence on the ratio D/d, in the w.h.p statements.



**Summary Of The Paper:**

The paper aims to provide a theoretical explanation for recent observations (lottery tickets, training in random subspaces, spanning pruning) that deep neural networks can be trained using fewer parameters than necessary. They provide the theoretical explanation using the so called Gordon escape theorem from high-dimensional geometry, which says that there is a phase transition in the success probability of training as the training dimension exceeds a threshold and this threshold is rather tight. This is supported by experiments on various benchmark datasets that seem to exhibit this phenomena of this phase transition.

**Summary Of The Review:**

In summary, the authors provide a theoretical explanation for many recent results that support training on a lower-dimensional subspace of the NN weight space and the back the theoretical claim via experimental results that show the existence of such phase transitions. Although, a lot remains to be said, this paper raises more interesting questions and I think this will be of value to the deep learning community.

---

> ### Author Response · Authors · 2021-11-20
> **Response to Reviewer fVom**
>
> **The theorem is a first such characterization, but it is not clear as to the difficulty of estimating the threshold dimensions for more than quadratic loss functions, potentially when involving deep ReLU networks for instance. It would be nice to have a discussion of the challenges for at least shallow networks (1 hidden layer ReLU network for example).**
>
> For general neural networks (even shallow ones), we do not expect to be able bound the local angular dimension; instead, we use the relationship between the threshold training dimension and local angular dimension to empirically probe this property of the loss landscape as in figure 3.  One reason this is challenging is that the loss landscape consists of many basins of local optimum connected by manifolds of low-loss points; see for example recent work on mode connectivity such as [1] and [2]. However, in our revision of the paper we have added supplement section A.1 that considers comparisons to networks linearized around a local optimum in the loss landscape via the Neural Tangent Kernel (NTK).  In these new experiments, we show that for Conv-2 trained on MNIST and Fashion MNIST, the threshold training dimension of the linearized model approximates the same property of the full model fairly well.  We thus think a promising direction is to theoretically characterize this linearized model (which has a single basin rather than the much more complicated "web" of solutions in full networks described above) as a reasonable approximation to the true landscape in the neighborhood of an optimum.
>
> [1] Stanislav Fort and Stanislaw Jastrzebski. "Large Scale Structure of Neural Network Loss Landscapes." NeurIPS 2019. https://proceedings.neurips.cc/paper/2019/hash/48042b1dae4950fef2bd2aafa0b971a1-Abstract.html
>
> [2] Gregory Benton, Wesley Maddox, Sanae Lotfi, Andrew Gordon Gordon Wilson. "Loss Surface Simplexes for Mode Connecting Volumes and Fast Ensembling." ICML 2021. https://proceedings.mlr.press/v139/benton21a.html
>
> **Another clarification is in Eq 2.2, the randomness of A is from a Gaussian distribution. In the statement of Gordon escape theorem it would be useful to clarify what "uniformly from a Grassmanian mean"?**
>
> In short, "uniformly from a Grassmanian" means in order to select a $d$-dimensional subspace of $\mathbb{R}^D$ we draw a $D \times d$ matrix whose entries are i.i.d. $\mathcal{N}(0, 1)$.  The column space of this matrix defines the subspace we have drawn.  This is now stated explicitly in the experiment supplement, Section A.4.
>
> The Grassmanian $G_{(D, d)}$ is the manifold that consists of all $d$-dimensional subspaces of $\mathbb{R}^D$.  If we make this a measure space $(G_{(D, d)}, \mathbb{P})$ using the uniform or Haar measure as $\mathbb{P}$, it can be shown that that the above procedure corresponds to sampling a random subspace $E \sim \text{Unif}(G_{(D, d)})$.
>
>
> **Are there any restrictions on how the random subspace is generated? As a continuation, in the Lottery subspaces algorithm, the matrix U is generated by the dynamics of GD on the loss function. Does the Gordon escape theorem still apply for this random process?**
>
> For our main theorem to hold (Theorem 3.2), the subspace must be drawn uniformly from the Grassmanian as described above.  To get similar results for another method generating a random subspace, one would first have to prove that a result akin to the Escape Theorem holds for this method.
>
> Importantly, this is *not* true for the lottery subspaces generated from the dynamics of GD. This method is intended to provide an empirical comparison in order to understand how much training can be used to improve the selection of a subspace.
>
> **Is it possible to infer in the pruning algorithm and the Gordon escape theorem, how the trainable network depends on depth or width of the original network? It would be interesting to see at least experimentally the change in depth/width for large classes of problems, as this ties in with numerous results on expressivity that say in general , increasing depth is more useful than increasing width (from the lens of expressivity alone!).**
>
> As the theory currently stands, the effects of depth/width would have to be probed via experiment. We do study three different sized models on each of the datasets considered, but did not set up the experiments to systemically study the effects of depth/width. We think that such experiments (alongside the question of how the properties of the dataset affect pruning across models) are important future work!
>
> **It would be useful if the authors can clarify the dependence on the ratio D/d, in the w.h.p statements.**
>
> Since our w.h.p. statements are explicitly in terms of $D$ and $d^*$,  they can be arranged to be in terms of this ratio.  For example, the threshold training dimension can be written as $d^*/D = 1 - d_{\text{local}}/D$.  We would be happy to include this form as well if the reviewer thinks it will make the paper clearer.

---

> > ### Comment · Reviewer_fVom · 2021-11-29
> > **Acknowledgement for reading the author response**
> >
> > Thank you for your detailed response to each question. After the responses, I think the authors have clarified most questions and I think the paper explores some interesting theoretical directions, that may guide the understanding of Deep neural networks, hopefully for more practical settings, ReLU functions and so on in the future. Thus I retain my current score.

---

### Official Review · Reviewer_krRB · 2021-11-02

**Correctness:** 4
**Technical Novelty And Significance:** 4
**Empirical Novelty And Significance:** 4
**Recommendation:** 8
**Confidence:** 4

**Main Review:**

Pros:

-- A new theoretical result that explains the existence of threshold training dimension.

-- The paper is well motivated and well structured. In particular, the authors showed in a toy example that the theoretical lower bound is close to the empirically observed value. The authors provided extensive experimental analysis of transition bounds for different random subspaces.

Cons:

-- The lottery subspaces, despite having smaller transition dimensions, encode a lot of information into the linear transformation matrix, therefore applications to compression can be limited.

-- The considered theory does not take into account optimization methods.

Questions and notes:

-- Figure 2, last column. The training threshold dimensions look similar for burn-in subspaces with different starting iterations. Therefore the argument that a better starting point decreases the threshold dimension does not look convincing. Does one burn-in step correspond to one batch update?

-- Section 2. Loss sublevel sets paragraph. l(f_w(x), y) -> l(f_w(x_n), y_n)



**Summary Of The Paper:**

The modern deep neural networks are over-parameterized and it is possible to construct a smaller parameter space that delivers a similar loss value. If one investigates the dependency of the probability of achieving the desired loss values on training subspace size, one will observe a sharp transition. The dimension where this transition happens the authors call a threshold training dimension. The authors of this paper proposed a theoretical explanation of the existence of the threshold training dimension. In particular, using Gordon’s escape theorem the authors describe the dependence of the threshold training dimension on the initialization and final desired loss. The authors proposed new lottery subspaces for which the threshold training dimension is much higher than for other random subspaces.

**Summary Of The Review:**

Overall, I would recommend accepting the paper. It contains interesting theoretical results explaining the existence of threshold training dimensions.

---

> ### Author Response · Authors · 2021-11-20
> **Response to Reviewer krRB**
>
> **The lottery subspaces, despite having smaller transition dimensions, encode a lot of information into the linear transformation matrix, therefore applications to compression can be limited.**
>
> Yes, we do not think these methods will be useful for compression.  Rather we see them primarily as scientific tools to explore loss landscapes and better understand the space of pruning methods.
>
> **The considered theory does not take into account optimization methods.**
>
> There are several places where optimization occurs in our methods: training within a chosen hyperplane, training for the burn-in epochs, and training to form a lottery subspace. We consider how improving the optimization method would affect our experiments in each case.
>
> 1. Training within a hyperplane: Here we are simply trying to determine whether or not an intersection occurs with a given sublevel set and because we cannot guarantee that we have found the lowest loss value in the hyperplane, we actually obtain an upper bound on the threshold training dimension. Improving the optimization method would simply affect the quality of this upper bound.
>
> 2. Training for the burn-in epochs: After the burn-in procedure is completed, what matters according to our theory is the distance to the sub-level set (or more precisely the extent of the sublevel set projected onto a sphere around this point).  Using a better optimization method could move you closer in a smaller number of training steps, but the qualitative behavior of burn-in should remain the same: as you increase the burn-in epochs, you move closer to the sublevel set and the training dimension decreases.
>
> 3. Training to form a lottery subspace: This is the situation where changing the optimization procedure would make the most difference because the training trajectory determines the basis for the subspace. In this situation, we think it would be interesting to compare across different optimization methods.
>
> **Figure 2, last column. The training threshold dimensions look similar for burn-in subspaces with different starting iterations. Therefore the argument that a better starting point decreases the threshold dimension does not look convincing. Does one burn-in step correspond to one batch update?**
>
> Yes, one burn-in step corresponds to one batch update (with a batch size of 128 images).  While we agree that for high accuracy the threshold training dimension becomes similar across burn-in experiments, in most dataset/architecture combinations the threshold training dimension increases with burn-in steps below a given accuracy (i.e. the red, green, orange, and blue curves do not lie on top of each other below a given accuracy).  We hope this is further reinforced by the experiments in Fig 3 which include the same type of experiment as the last column of Figure 2 but for additional models (Conv-3 and ResNet-20).
>
> **Section 2. Loss sublevel sets paragraph. $l(f_w(x), y) -> l(f_w(x_n), y_n)$**
>
> This has been corrected.  We thank the reviewer for catching this error.

---

> > ### Comment · Reviewer_krRB · 2021-11-24
> > **Comment**
> >
> > I would like to thank the authors for their reply. Based on other reviews and the authors' replies, I believe it is a good paper and I would like to leave my score unchanged.

---

### Official Review · Reviewer_17kR · 2021-11-02

**Correctness:** 3
**Technical Novelty And Significance:** 3
**Empirical Novelty And Significance:** 3
**Recommendation:** 6
**Confidence:** 4

**Main Review:**

Gordon's escape theorem characterizes the probability that a given subset of the unit sphere $S\in \mathbb{S}^{D-1}$ has an empty intersection with a random $d$-dimensional subspace of $\mathbb{R}^D$. The randomness here is given by a standard Gaussian matrix and the random subspace is distributed uniformly on the Grassmann manifold w.r.t. the associated Haar measure. The probability of escape (i.e., no intersection) is characterized by a squared version of the Gaussian width of $S$. The Gaussian width is the expected value of the dual norm of a standard Gaussian vector, and for sets containing the origin it coincides with the more familiar notion of Gaussian complexity in statistical learning theory.

The innovation in this paper is a straightforward adaptation of Gordon's result for estimating the success probability of hitting the convex hull of a loss sublevel set when training within a random subspace of the original parameter space. For the success probability to be large, the subspace codimension $D-d$ must be less than the squared Gaussian width of the desired training loss sublevel set projected onto a sphere around the initialization. The authors identify the latter as the critical threshold training codimension in relation to the observations of Li et al (2018) and empirically demonstrate a phase transition in the success probability in the loss-subspace dimension plane around this threshold. For a quadratic approximation of the loss, the authors give a bound on the threshold dimension in terms of the spectrum of the Hessian and distance to initialization.

The main strength of the paper lies in exploring the ramifications of the above for training within random and optimized subspaces starting from either a random initialization or after using some information from training the network in the full space (which the authors call "burn-in") that apparently brings the initialization closer to a desired loss sublevel set. This explains for instance why the threshold dimension decreases when more information is burned into the initialization.

Question:

Can you comment on the estimation of the threshold dimension for general losses (beyond the quadratic approximation)?

Minor nitpicking:

-- The reference to Gordon's theorem both in Gordon (1988) which I believe is Corollary 3.4 and also in the Mixon (2014) blog state the result with coefficient for the exponential as 3.5 instead of 2.5. I can see how one gets the coefficient 2.5, but perhaps for the benefit of the reader, the authors should furnish a reference that accurately reflects their statement of Gordon's theorem in relation to the above.

-- In the second paragraph in page 1, the statement should be "... this threshold training dimension is equal to the dimension of the full parameter space minus the "squared" Gaussian width of the desired loss sublevel set projected onto the unit sphere around initialization." (include the squared?).


**Summary Of The Paper:**

This paper builds on the empirical observation of Li et al (2018) that training neural networks in a low-dimensional random subspace of the parameter space often achieves similar levels of train and test accuracies as training in the original parameter space when the subspace dimension is above a critical threshold. The main result is an application of Gordon's escape theorem showing that the critical threshold training codimension is the squared Gaussian width of loss sublevel sets projected onto a sphere around the initialization.

**Summary Of The Review:**

The contribution is a straightforward application of Gordon's result weighing mainly on the empirical side. I believe nonetheless that the paper gives new insights for training in random subspaces. Overall, I like the paper and would be happy to raise my score based on the authors' response.

---

> ### Author Response · Authors · 2021-11-20
> **Response to Reviewer 17kR**
>
> **Can you comment on the estimation of the threshold dimension for general losses (beyond the quadratic approximation)?**
>
> For general neural networks, we do not expect to be able bound the local angular dimension; instead, we use the relationship between the threshold training dimension and local angular dimension to empirically probe this property of the loss landscape as in figure 3.  However, in our revision of the paper we have added a section to the supplement (Section A.1) that considers comparisons to networks linearized around a local optimum in the loss landscape via the Neural Tangent Kernel (NTK).  In these new experiments, we show that for Conv-2 trained on MNIST and Fashion MNIST, the threshold training dimension of the linearized model approximates the same property of the full model fairly well.  We thus think a promising direction is to theoretically characterize this linearized model (which has a single basin rather than the much more complicated "web" of solutions in full networks) as a reasonable approximation to the true landscape in the neighborhood of a given optimum.
>
> **The reference to Gordon's theorem both in Gordon (1988) which I believe is Corollary 3.4 and also in the Mixon (2014) blog state the result with coefficient for the exponential as 3.5 instead of 2.5. I can see how one gets the coefficient 2.5, but perhaps for the benefit of the reader, the authors should furnish a reference that accurately reflects their statement of Gordon's theorem in relation to the above.**
>
> As it does not impact our results, we have changed this coefficient to 3.5 instead of 2.5 to match Mixon (2014) and the original paper by Gordon.  We thank the reviewer for pointing out this inconsistency.
>
> **In the second paragraph in page 1, the statement should be "... this threshold training dimension is equal to the dimension of the full parameter space minus the "squared" Gaussian width of the desired loss sublevel set projected onto the unit sphere around initialization." (include the squared?).**
>
> Yes, this has been corrected.  Thank you for catching this!

---

> > ### Comment · Reviewer_17kR · 2021-11-24
> > **Response to authors**
> >
> > Thanks for your response. You can find a reference to Gordon's result (with the factor 2.5 in your original submission) in Theorem 4.3 here:
> >
> > Mark Rudelson, Roman Vershynin (2007), "On sparse reconstruction from Fourier and Gaussian measurements" https://doi.org/10.1002/cpa.20227

---

### Decision · Program_Chairs · 2022-01-20

**Decision:**

Accept (Poster)

**Comment:**

This paper presents new insights for training on random subspaces of low dimension, with several theoretical and experimental contributions. This is a paper that would be interesting to many people doing research in deep learning, both from the theoretical and practical side.